# Rights based approaches to sexual and reproductive health in low and middle-income countries: A systematic review

Majel McGranahan[1]*, Joselyn Nakyeyune[2], Christopher Baguma[2], Nakibuuka Noor Musisi[2], Derrick Nsibirwa[2], Sharifah Sekalala[3], Oyinlola Oyebode[1]

1 University of Warwick Medical School, Coventry, West Midlands, United Kingdom, 2 Center for Health, Human Rights and Development (CEHURD), Kampala, Uganda, 3 University of Warwick, Law School, Coventry, West Midlands, United Kingdom

* majel.mcgranahan@warwick.ac.uk

**Data Availability Statement:** All relevant data are within the manuscript and its Supporting information files.

## Abstract

### Introduction

The Sustainable Development Goals, which are grounded in human rights, involve empowering women and girls and ensuring that everyone can access sexual and reproductive health and rights (Goal 5). This is the first systematic review reporting interventions involving rights-based approaches for sexual and reproductive health issues including gender-based violence, maternity, HIV and sexually transmitted infections in low and middle-income countries.

### Aims

To describe the evidence on rights-based approaches to sexual and reproductive health in low and middle-income countries.

### Methods

EMBASE, MEDLINE and Web of Science were searched until 9/1/2020. Inclusion criteria were:

- Study design: any interventional study.

- Population: females aged over 15 living in low and middle-income countries.

- Intervention: a "rights-based approach" (defined by the author) and/or interventions that the author explicitly stated related to "rights".

- Comparator: clusters in which no intervention or fewer components of an intervention were in place, or individuals not exposed to interventions, or exposed to fewer intervention components.

- Outcome: Sexual and reproductive health related outcomes.

A narrative synthesis of included studies was undertaken, and outcomes mapped to identify evidence gaps.

**Funding:** The University of Warwick supported this work through a Global Challenges Research Fund Fellowship and the Institutional Research Support Fund. MM is supported through Health Education England. OO is supported by the National Institute for Health Research Global Health Research Unit on Improving Health in Slums. Views expressed are those of the authors and not the funders.

**Competing interests:** The authors have declared that no competing interests exist.

The systematic review protocol was registered on PROSPERO (CRD42019158950).

## Results

Database searching identified 17,212 records, and 13,404 studies remained after de-duplication. Twenty-four studies were included after title and abstract, full-text and reference-list screening by two authors independently.

Rights-based interventions were effective for some included outcomes, but evidence was of poor quality. Testing uptake for HIV and/or other sexually transmitted infections, condom use, and awareness of rights improved with intervention, but all relevant studies were at high, critical or serious risk of bias. No study included gender-based violence outcomes.

## Conclusion

Considerable risk of bias in all studies means results must be interpreted with caution. High-quality controlled studies are needed urgently in this area.

## Introduction

An estimated 810 women died every day in 2017 from pregnancy or childbirth-related preventable causes; 94% of these deaths were in low and middle-income countries (LMICs) [1]. But it is not just childbirth where women suffer poor sexual and reproductive health outcomes: women and girls make up nearly three quarters of new infections of HIV worldwide amongst those aged 10–19 [2]. Gender-based violence increases the risk of HIV [3], and in some areas, up to 45% of girls' first sexual encounter was forced [4]. Between 2010 and 2018, there was a reduction in new HIV infections in females between 15–24 years but despite this, there are approximately 6000 new infections per week worldwide [5]. The Joint United Nations Programme on HIV/AIDS (UNAIDS) recommends that to combat HIV we must advance gender equality and human rights [6].

By 2030, the Sustainable Development Goals (SDG), which are grounded in human rights, include empowering women and girls (Goal 5) and ensuring "universal access to sexual and reproductive health and reproductive rights" (Section 5.6) [7]. The United Nations Population Fund (UNFPA) define good sexual and reproductive health as "a state of complete physical, mental and social well-being in all matters relating to the reproductive system. It implies that people are able to have a satisfying and safe sex life, the capability to reproduce, and the freedom to decide if, when, and how often to do so" [8].

The International Covenant on Economic, Social and Cultural Rights, which opened in 1966 and came into force in 1976, requires States which are under the Charter of the United Nations to promote human rights [9]. The UN Common Understanding on a Human-Rights-Based Approach to Development Cooperation outlines what a human rights based approach should entail (as a minimum) [10]. Essential elements include accounting for the recommendations of international human rights bodies; monitoring and evaluating outcomes and processes following human rights principles and standards; identifying barriers and building capacity for individuals (rights holders) to claim their rights and for duty-bearers' to fulfil their obligations to support rights holders to claim their rights (including identifying what the rights holders' human rights claims are) [10]. This systematic review will focus on interventions that primarily address a rights-based approach from the rights holder's perspective (rather than

those of duty-bearers) in order to include a manageable number of papers and meaningful synthesis.

An effective rights-based approach should enable women to actively take part in decisions regarding their sexual and reproductive health, and challenge those who are preventing them from doing so [11]. The Office of the United Nations High Commissioner for Human Rights (OHCHR) Technical Guidance on rights-based approaches to prevent maternal morbidity and mortality describes a rights-based approach as "premised upon empowering women to claim their rights" [11]. This requires mechanisms for accountability for the vindication of rights, explicit recognition of a woman's right to health (which includes a thorough understanding of sexual and reproductive health in legislation and and/or constitutions), and a focus on health rather than individual pathologies [11].

Previous reviews of rights-based approaches have been limited to certain aspects of sexual and reproductive health. A systematic review, published in 2015, looked at interventions that aimed to promote awareness of rights to increase use of maternity services; results from the four included studies indicated that interventions resulted in an increase in some aspects of service-use, including antenatal care, but studies were of varying quality [12]. The review was limited to maternity services and only included studies that explicitly included awareness-raising (excluding others even if they used rights-based approaches). A more recent systematic review assessed the impact of five aspects of the Joint United Nations Programme on HIV/AIDS (UNAIDS) categories of human rights programs on HIV-related outcomes between 2003–2015 (human rights and medical ethics training; legal literacy; legal services (HIV-related); reforming and monitoring regulations, laws and policies related to HIV; law-maker and law enforcement agent sensitisation) [13]. As well as adopting a limited definition of a rights-based approach, the study only focused on HIV-related outcomes. A (non-systematic) literature review of human rights-based approaches to women's and children's health found that rights-based approaches were linked to improved health-related outcomes but was limited to studies using a "participatory approach" [14].

Thus, no systematic review has described the evidence on rights-based approaches to sexual and reproductive health more broadly, including non-HIV and maternity related outcomes such as gender-based violence and other sexually transmitted infections (STI). Policy-makers, governments and health services require evidence on rights-based approaches to sexual and reproductive health to inform the development of interventions and provision of services for their populations. This systematic review examines the evidence on explicitly rights-based approaches to sexual and reproductive health among women living in LMICs.

## Methods

The systematic review protocol was registered with PROSPERO on 4/12/2019 (CRD42019158950) [15]. The Preferred Reporting Items for Systematic Reviews and Meta-Analyses (PRISMA) were followed, as described in the PRISMA flow diagram (Fig 1) and checklist (S1 File).

EMBASE, MEDLINE and Web of Science were searched from inception until 9/1/2020. Search terms included medical subject headings or equivalent and free text terms including sexual health, gender-based violence, maternal health, human rights and more, combined using Boolean operators (S2 File). Search terms were chosen to cover the subject areas of contraception, pregnancy, STIs, awareness of rights, violence and mental health. Reference lists of included studies and relevant identified systematic reviews were screened to identify any further studies not identified in the initial search. There were no language restrictions. Reviews of published studies in languages other than English were conducted by the authors themselves

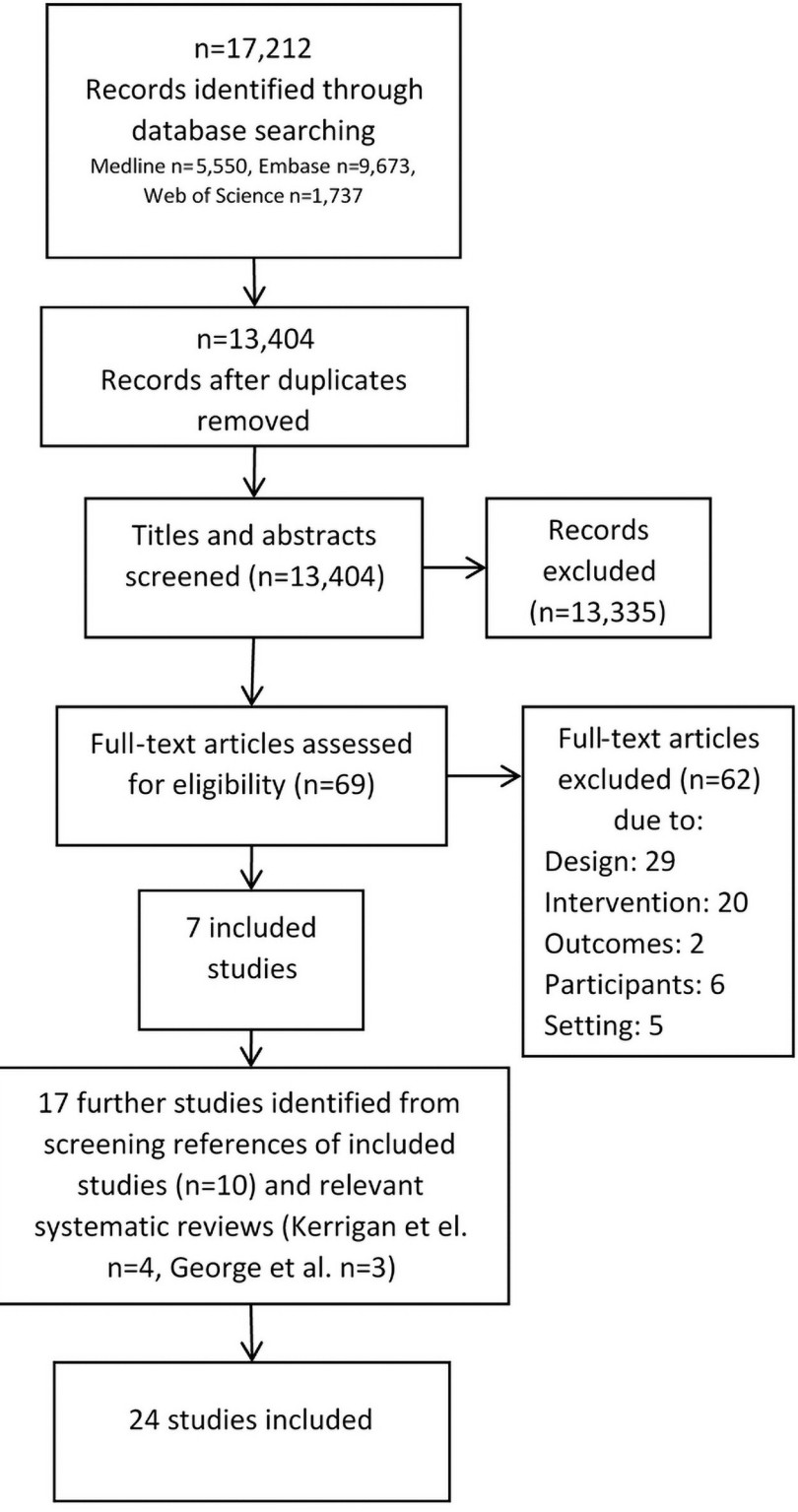

**Fig 1. PRISMA flow diagram.**

(French, Portuguese) or a University of Warwick colleague, Dr Yen-Fu Chen, (Mandarin). The search strategy was reviewed by a specialist academic librarian. Searches were adapted for each database. Grey literature was not included due to the anticipated lack of detail in the methods of published studies, and anticipated lack of independence of published evaluations (for example, evaluations published by those running an intervention as part of seeking further funding for their organisation or intervention).

## Inclusion criteria

- Study design: any interventional study, including randomised controlled trials (RCTs), cohort studies, case-control studies and before-and-after studies.

- Population: females aged over 15 years living in LMICs (defined by the World Bank Group's classification [16]).

- Intervention: any approach the author described as a "rights-based approach", or involving sexual and reproductive health interventions that authors explicitly related to "rights" (for example, an intervention that promoted the awareness of rights). We also included studies based on an intervention referred to as rights-based in another included paper. We only included interventions directed towards women (other than healthcare professionals) or their families. Studies using educational interventions were included in addition to non-educational interventions, as long as they were explicitly described by study authors as related to "rights".

- Comparator: Acceptable comparators included clusters in which no intervention or fewer components of an intervention (for example without the rights-based components) were in place, or individuals not exposed to interventions, or exposed to fewer intervention components.

- Primary outcomes, measured through surveys or routine data (health or crime):

  - Gender-based violence (prevalence/incidence)

  - STIs (testing/prevalence/incidence)

  - Pregnancy and delivery-related outcomes (including antenatal, perinatal and post-partum healthcare access, pregnancy/birth complications)

  - Perinatal mental health (incidence/prevalence)

  - Unintended pregnancy (incidence)

  - Female genital mutilation (incidence/prevalence)

  - Contraceptive/condom use

  - Sexual and reproductive healthcare attendance (number of visits)

  - HIV-related outcomes (including viral load, CD4 count, medication adherence)

  - Mental wellbeing as measured by any wellbeing scale

  - Awareness of rights

- Secondary outcomes, measured through surveys or routine data (health or crime):

  - Healthcare attendance unrelated to sexual and reproductive health (number of visits)

- Healthcare workers' understanding of rights (difference between before and after intervention)

### Exclusion criteria

- Study design: Qualitative studies

- Population: Children under 15 years. People who are male or transgender.

Following database searches, titles and abstracts were screened by two reviewers independently. Each included full-text article was then reviewed by two reviewers independently. At each state, if there *were* discrepancies regarding inclusion/exclusion decisions, they were addressed initially via discussion, and if necessary via a third author.

Rayyan Systematic Review Web App [17] and Microsoft Excel were used to record decisions.

Data were extracted by two reviewers independently onto a data extraction tool developed by the authors which included basic study details (journal, dates, authors, and location, sample size, population, study design, study duration, interventions, statistical analysis, results, ethics and funding). Only relevant results pertaining to populations eligible for inclusion in this systematic review were extracted- for example, where data was collected for males and females, only data presented on females were extracted.

### Analysis

A narrative synthesis of included studies was undertaken. Due to study heterogeneity, meta-analysis was not undertaken. Outcomes were mapped to identify evidence gaps. Effectiveness of identified interventions was examined based on the author's choice of outcomes.

Quality and risk of bias was assessed by two reviewers independently using the RoB-2 tool for cluster-randomised studies [18], and the ROBINS-I tool for non-randomised studies [19]. Uncontrolled before-and-after studies based on two cross-sectional surveys were given an overall assessment of serious or critical risk of bias [20]. Disagreements were addressed via discussion and if necessary via a third author.

### Risk of bias in included studies

All studies were classed as high, serious or critical risk of bias, meaning that due to issues with the quality of the studies, there was a substantial risk that included studies overestimated or underestimated the true effect of the interventions. The risk of bias in cluster RCTs and non-randomised interventional studies is displayed in S1 and S2 Tables respectively. Uncontrolled cross-sectional before-and-after studies were all classified as serious or critical risk of bias (S3 Table).

### Results

Of 17,212 records identified through database searching, 13,404 remained after de-duplication (Fig 1), 69 remained after title and abstract screening, and seven remained after full-text screening [21–27]. Reference list screening of relevant systematic reviews [12,13] identified a further seven [28–34], and screening of included studies identified ten further studies [35–44]. Twenty-four studies were included in the final systematic review, twenty-three in English and one in Portuguese. Outcomes identified in the 24 studies were: condom use; HIV and other

STIs; testing for HIV and other STIs; knowledge of STIs and the prevention of STIs; intimate partner violence; awareness of rights; healthcare attendance; and healthcare access and/or interventions during pregnancy and delivery.

## Population and setting

As shown in Table 1, thirteen studies were undertaken among female sex workers (FSWs) in India: eight in south India as part of the evaluation of the India AIDS initiative, Avahan [23,31–33,35–38], and five in West Bengal, India, as part of the Sonagachi project [24,39–41,44].

The remaining eleven studies were undertaken in Bangladesh (married women living in slums) [21], Philippines (FSWs) [22], Tanzania (women in the third trimester of pregnancy) [25], East Nepal (women who had delivered a baby within three years) [26], India (pregnant women [28] and women who had delivered a baby within twelve months [30]), Uganda (women living in rural areas) [29], Kenya (women who had given birth in last 24–48 hours) [42], the Amazon region of Brazil (FSWs- two papers using the same dataset) [34,43], and Egypt (women who were newly married, pregnant or post-partum) [27].

## Study design and interventions

Four studies were cluster RCTs [21,28,29,40], two non-randomised interventional studies [24,39], one a time-series study design [23], and seventeen were retrospective observational studies based on one or more cross-sectional surveys [22,23,25–27,30–38,41–44].

All studies involved multiple components, including various elements alongside (or part of) a rights-based approach. These included education/workshops on rights and sexual health, sexual health service provision, capacity building, advocacy, community mobilisation, peer support and outreach, condom access and promotion, leaflets and posters, and community meetings (Table 1).

## Outcomes and results

Table 2 shows the results of each study. Both Benzaken et al. 2003 and 2007 [34,43], and Jana et al. 1995 and 1998 [41,44] contained data from the same study so are reported together.

**Condom use.** Eleven studies showed a significant positive association between intervention exposure (before-versus-after, self-reported exposure or intervention-versus-control group) and condom use for some [22,31,33–35,37,40,43] or all [24,32,36] of the relevant study population; all of these studies were at serious, high or critical risk of bias. One study, at serious risk of bias, found no significant association [39].

**HIV.** Ramesh et al. reported a significant reduction in HIV [35], whereas Reza-Paul et al. [36] reported no significant change. Jana et al. [44] reported increased HIV prevalence rates in three consecutive years after intervention implementation, but no statistical significance level was given. All three studies were at critical risk of bias.

**Other STIs.** Jana et al. reported a non-significant reduction in syphilis [44]. Other studies showed different effects for different STIs [35,36], and Gangopadhyay et al. found no significant effect on lab-proven STI rates [39]. Halli et al. reported increased STI symptoms among those with higher self-reported intervention exposure [37]. All five of these studies had a critical [35,36,44], or serious risk of bias [37,39].

**Testing for HIV/STIs.** Urada et al. (critical risk of bias) found intention to test for HIV was significantly higher post-intervention [22]. Gangopadhyay et al. (serious risk of bias) reported a significantly increased proportion of FSWs having regular STI check-ups and HIV

**Table 1. Characteristics of included studies.**

| Author and year | Population/setting (inclusion/exclusion, age, number) | Intervention (including comparator) | Outcomes |
|---|---|---|---|
| | | *Studies focusing on maternity* | |
| **Metwally et al. 2019** [27] | *Setting*: Egypt: 21 villages and 119 satellites in Al Fayoum and Benisuef, 11 villages and 76 satellites in Senores and Youssef El Sedeek districts and 10 villages and 43 satellites in El Fashen and Somosta districts. *Population*: Newly married, post-partum or pregnant women of reproductive age. Pre-intervention: 1000 women Post-intervention: 1063. No ages given. | 1. Education and promotional materials including posters, a calendar, a game and a drama production on CD which included a message regarding pregnancy-related rights. Information on service availability and what to do in emergencies was also included. 2. Training for health workers 3. Outreach and education sessions *Comparator*: baseline Referred to rights | Awareness of rights within healthcare including the right to have 1. Follow up with a trained health professional 2. Education sessions at the health centre 3. Delivery in a healthcare facility |
| **Ratcliffe et al. 2016** [25] | *Setting*: Large referral hospital in Dar es Salaam, Tanzania *Population*: Women in their third trimester of pregnancy attending antenatal care at the study facility between May-October 2014. All 362 invited women accepted. Ages not specified of those doing pre-and post-intervention Open Birth Day test. | 1. Open Birth Days for women in their third trimester, including education about antenatal care and preparing for birth, improving knowledge of patient rights and empowerment of women to advocate for their health care rights during childbirth 2. Posters of the 'Universal Rights of Childbearing Women' were displayed in all maternity wards 3. Respectful Maternity Care Workshops also provided for healthcare providers. *Comparator*: Pre-session test Referred to rights | Awareness of rights with respect to healthcare received during birth: 1. Due care and attention 2. Consent for procedures 3. Not being shouted at 4. Not being physically assaulted 5. Privacy within hospital 6. Discrimination based on religion/age/ethnicity/wealth 7. Leaving the hospital 8. Asking questions (Immediately before and after intervention) |
| **Abuya et al. 2015** [42] | *Setting*: 13 Kenyan health facilities in a mixture of urban, peri-urban and rural areas including one slum. A mixture of private, public and faith-based hospitals and a (public) health centre. *Population*: women aged 15–45 years who had given birth in the previous 24–48 hours in a participating facility, regardless of pregnancy outcome. Baseline: 641 women, end-line: 728 Mean age at baseline 25.0 (5.3) and end-line 25.2 (5.1) | 1. Policy-level: policy dialogue with professional networks, government and civil society 2: Facility-level: training for providers and managers on respectful maternity care 3: Community level: community workshops including education regarding rights to sexual and reproductive health including maternal health (for both females and males). Training on conflict resolution and counselling for those who had experienced disrespect and abuse *Comparator*: baseline Referred to rights-based approach–indicates this used although not explicitly | Outcomes during labour and delivery or examination: 1. Confidentiality violation 2. Feeling humiliated or disrespected 3. Lack of privacy 4. Privacy violation 5. Physical abuse/aggression 6. Detention 7. Verbal abuse/aggression 8. Abandonment 9. Non-consented care |
| **Rana et al. 2012** [26] | *Setting*: eight districts in East Nepal *Population*: delivery outcomes collected from participants who delivered a baby in the previous 3 years (181 females, ages not given). Full questionnaires undertaken with 320 households in intervention group and 320 households in control groups | 1. Rights-based approach promoted by local NGOs 2. Concepts of health rights introduced to groups considered disadvantaged. 3. Health rights advocacy and improved healthcare was promoted at national, regional and village levels 4. Health promotion, street theatre and youth mobilisation and support for local health committees *Comparator*: Similar households in villages with no intervention Rights-based approach | 1. Place of delivery (home or healthcare institution) 2. Skilled birth attendant present at delivery 3. Use of "clean home delivery kit" (if applicable, yes or no) (Measured by structured interview) |

**Table 1.** (Continued)

| Author and year | Population/setting (inclusion/exclusion, age, number) | Intervention (including comparator) | Outcomes |
|---|---|---|---|
| **Björkman et al. 2009** [29] | *Setting*: primary health-care providers across rural Uganda<br>*Population*: 5,000 households surveyed in each round (stratified random sample within catchment area of each facility, out of approximately 55,000 households across nine districts) | Community based monitoring intervention including<br>1. Posters on patient rights<br>2. Report cards outlining service availability and comparisons between areas<br>3. Information on patients' rights and entitlements<br>4. Meetings between community and health workers to discuss responsibilities and rights.<br>*Comparator*: No intervention<br>Clusters randomly assigned<br>Referred to rights | 1. Number seeking antenatal care<br>2. Number seeking family planning<br>3. Number of pregnancies<br>(Surveys implemented prior to intervention and one year after project initiation) |
| **Sinha 2008** [30] | *Setting*: Mominpet Mandal within Rangareddy district of Andhra Pradesh, rural India<br>*Population*: Women who had delivered a baby in previous 12 months. Baseline—random sample, end-line—all women. Ages not given. | 1. Discussions with the community, raising awareness of women's rights including the right to access healthcare<br>2. Posters on pregnant women's rights to services, the importance of non-home delivery and information regarding the legal minimum age for marriage<br>3. Training manuals<br>4. Regular meetings<br>5. Visits to pregnant women, support with access to healthcare and certain medications<br>*Comparator*: baseline<br>Intervention "based on a rights approach" | 1. At least one antenatal check-up<br>2. More than three antenatal visits<br>3. Received antenatal care during first trimester<br>4. Given 100 or more iron and folic acid tablets<br>5. At least two tetanus toxoid injections given<br>6. Decision regarding place of delivery (home vs. institution)<br>7. Actual place of delivery (home, primary health centre, government hospital, private clinic/hospital)<br>(End-line included questions regarding experience of intervention and penultimate pregnancy) |
| **Pandey et al. 2007** [28] | *Setting*: Uttar Pradesh, India<br>*Population*: 105 village clusters (average population of 2343 per cluster) across 21 districts.<br>Intervention group: 144 households with pregnant women at baseline, and 168 at end line<br>Control group: 105 pregnant women at baseline, and 102 at end line | Information campaign which consisted of:<br>1. Leaflets and posters<br>2. 4–6 meetings<br>3. Distribution of information regarding villagers' rights to services and the process of making complaints<br>*Comparator*: no intervention<br>Referred to rights | 1. Proportion of pregnant women who:<br>2. Were visited by a nurse midwife<br>3. Received a prenatal examination<br>4. Received prenatal supplements<br>5. Received the tetanus vaccine |
| | | *Studies focusing on violence*: | |
| **Naved et al. 2018** [21] | *Setting*: 19 slums in Dhaka, Bangladesh<br>*Population*: Data collected for 2,666 married women at baseline and 2,670 at follow-up, aged 15–29. | 1. Single-gender interactive sessions on health, gender, rights and life skills (+community mobilization and services)<br>*Comparator*: community mobilization and services only<br>Randomly assigned at cluster level.<br>Referred to rights | Experience of<br>1. Physical violence<br>2. Sexual violence<br>3. Emotional violence<br>(At 4 and 24 months post intervention) |
| **Beattie et al. 2010** [38] | *Setting*: Karnataka state, south India<br>*Population*: 3,852 FSWs took part in Integrated behavioural and biological assessments (IBBAs) (1,882 at baseline and 1,970 at follow-up, mean age 31.7) and 7,638 took part in polling booth surveys (PBS). | \*\*Avahan\*\*<br>1. Legal empowerment workshops<br>24-hour crisis management teams<br>Emergency contact if wrongfully arrested, sexually assaulted or suffered from violent attack<br>2. Legal literacy for sex workers provided by volunteer human rights lawyers<br>3. Training for police officers<br>4. Awareness raising for journalists<br>5. Advocacy at policy-level<br>*Comparator*: baseline (early on in intervention)<br>Referred to rights | IBBA data: Raped or beaten in past year by exposure to<br>1. Peer educator<br>2. Drop-in centre<br>3. Project sexual health clinic<br>4. Project "grey pack" or<br>5. Witnessing a condom demonstration<br>(Measured 12–16 months after programs commenced and again 33–37 months later).<br>PBS data:<br>1. Change in proportion experiencing violence over time from 2006 to 2007 to 2008 |

*(Continued)*

**Table 1.** (Continued)

| Author and year | Population/setting (inclusion/exclusion, age, number) | Intervention (including comparator) | Outcomes |
|---|---|---|---|
| | *Studies focusing on HIV/STIs among FSWs:* | | |
| **Urada et al. 2016** [22] | *Setting*: Cubao in Quezon City, Manila, Philippines<br>*Population*: Sex workers engaged in commercial sex in past 6 months. Venue-based females median age 23 (IQR 21–26). Street-based females median age 25 (IQR 21–35). 59 females (and 37 males). | 4-hour intervention which included:<br>1. HIV and STI education<br>2. Education on human rights legislation including reproductive health rights and Philippine laws. Raising awareness of the laws that protect individual's rights against discrimination and abuse<br>*Comparator*: Same group before the intervention<br>Human rights focused intervention | Knowledge regarding:<br>1. Reproductive health rights<br>2. Human rights<br>3. Ethical rights for research participants<br>4. HIV and other STIs including transmission<br>5. Consistent condom use intentions<br>6. HIV testing intentions<br>7. HIV treatment intentions (if tested HIV positive) |
| **Erausquin et al. 2012** [32] | *Setting*: Rajahmundry, Andhra Pradesh, southern India<br>*Population*: 2276 FSWs (aged over 18 years who received money for sex at least once in past 12 months)<br>Mean age 32.5 (SD 8.21) | **Avahan**<br>Community mobilisation intervention including<br>1. Education by peer health educators<br>2. Work with police to improve FSW treatment by police<br>3. Availability of condoms, and STI testing and treatment<br>*Comparator*: No/lower exposure (self-reported)<br>Evaluating same intervention (Avahan) referred to as rights-based by Reza-Paul et al. | Consistent condom use in past 7 days with<br>1. Non-clients<br>2. All clients<br>3. Regular clients<br>4. Occasional clients<br>(Measured at three time-points: 2006, 2007 and 2009–2010) |
| **Guha et al. 2012** [33] | *Setting*: Tamil Nadu and Maharashtra, southern India<br>*Population*: 2032 FSWs in Tamil Nadu State and 2525 in Maharashtra State | **Avahan** or Avahan-like interventions including<br>1. Participated in meeting/training organised by Non-Governmental Organisation (NGO)<br>2. Member of a self-help group supported by NGO<br>3. Contacted by a peer educator<br>4. Member of a sex worker collective (considered separate from Avahan programmes)<br>*Comparator*: No intervention/not a member of group (self-reported)<br>Referred to rights | 1. Consistent condom-use with all clients for meeting/training group and for self-help group by location (Chennai/Mumbai/rest of Tamil Nadu/rest of Maharashtra) |
| **Deering et al. 2011** [31] | *Setting*: three districts in Karnataka state, southern India<br>*Population*: 775 FSWs, median age 30 (IQR 25–35) | **Avahan**<br>1. Peer outreach, increasing awareness of condom-use and how to negotiate<br>2. Increasing availability of STI testing and treatment, and condom access<br>3. Community mobilisation<br>*Comparator*: self-reported lack of exposure<br>Related to rights (not explicitly): intervention involved legal empowerment. Evaluating same intervention as other studies (Avahan) referred to as rights-based by Reza-Paul | Association between intervention exposure (self-reported) and odds of:<br>1. Condom use (last day worked, all clients)<br>2. Consistent condom use (occasional clients)<br>3. Consistent condom use (most recent repeat client) |
| **Gurnani et al. 2011** [23] | *Setting*: 20 districts in Karnataka, southern India<br>*Population*: FSWs (n = 51,171 regular contact with intervention, >100,000 at least one contact). Ages not given. | **Avahan**<br>Multi-component:<br>1. Legal empowerment workshops with FSWs<br>2. Establishment of drop-in centres and sexual health clinics<br>3. Advocating with government for FSW rights<br>4. Supporting FSWs to be part of district AIDs committees<br>5. Media advocacy<br>*Comparator*: baseline (early on in intervention)<br>Referred to rights | 1. Number of FSWs visiting sexual health clinic<br>2. Rights violations reported to police<br>(Results given are total number reported rather than change so cannot measure effect) |

*(Continued)*

**Table 1.** (Continued)

| Author and year | Population/setting (inclusion/exclusion, age, number) | Intervention (including comparator) | Outcomes |
|---|---|---|---|
| **Ramesh et al. 2010** [35] | *Setting*: 5 districts (Mysore, Belgaum, Shimoga, Bellary and Bangalore Urban) within Karnataka state, southern India<br>*Population*: FSWs, median age 30 (IQR 19–41). 2312 at baseline, 2400 at follow-up. | **Avahan**:<br>1. Outreach, condom promotion and STI screening<br>2. Sexual health services<br>3. Advocacy with stakeholders<br>4. Drop in centres<br>5. Capacity building and community mobilisation<br>*Comparator*: baseline (early on in intervention)<br>Evaluating same intervention as other studies (Avahan) referred to as rights-based by Reza-Paul | 1. Condom-use<br>2. Prevalence of HIV, syphilis (and high titre syphilis), chlamydia, gonorrhoea and trichomonas (the latter in Mysore only) |
| **Swendeman et al. 2009** [24] | *Setting*: Two rural towns in West Bengal, India.<br>*Population*: FSWs. 100 from intervention town and 100 from control town, randomly selected from the 350 FSWs each community<br>Mean age 27 (SD = 7, range 18–50) | **Sonagachi**<br>1. STI clinics including STI treatment, condom promotion and peer education<br>2. Empowerment including rights-based framing, micro-finance, community mobilisation and advocacy<br>*Comparator*: only received #1 (STI clinics)<br>Programme used a "rights-based frame" | 1. HIV/STI knowledge<br>2. Workplace autonomy and sexual negotiation skills<br>3. FSW motivation for change "frame"<br>4. Social support<br>5. Financial security<br>6. Political participation |
| **Reza-Paul et al. 2008** [36] | *Setting*: Mysore city, Karnataka state, southern India<br>*Population*: FSWs. Median age 30 years. 429 at baseline, 425 at follow-up. | **Avahan**<br>A "rights-based approach" including:<br>1. Community mobilisation<br>2. Peer-mediated outreach<br>3. Sexual health service access<br>4. Enhanced enabling environment<br>*Comparator*: baseline<br>Rights based approach | 1. Ever attended project STI clinic<br>2. Condom breakage in past month<br>3. Unprotected sex<br>4. Condom use with commercial partner/regular partner<br>5. HIV-1, HSV-2, syphilis, high-titre syphilis, trichomonas, chlamydia and gonorrhoea infection |
| **Benzaken et al. 2007** [34] **and Benzaken et al. 2003** [43] | *Setting*: Town of Manacapuru in the Amazon region of Brazil (only accessible via riverboat)<br>*Population*: Probabilistic sample obtained from 500 FSWs, stratified by meeting points. Mean age of 25 (range 11–63) in 1999 and 24.2 (range 13–59) in 2001. | Five FSWs trained to become peer educators; their work included<br>1. Condom promotion<br>2. Education about STIs and AIDS<br>3. Referral of FSWs with suspected STIs to services<br>4. Reselling low-cost condoms. Training programme for healthcare workers on a syndromic approach to sexual health and counselling.<br>In addition, seminars on women's rights were held on International Women's Day (8th March) in 1999, 2000 and 2001<br>*Comparator*: baseline<br>Referred to rights | 1. Sexual behaviour<br>2. HIV testing<br>3. Condom use (for each of: anal, oral and vaginal sex, regular partners, clients, last week, all situations)<br>4. Reduction in number of sexual partners<br>5. Symptoms of STIs (ulcers, vaginal discharge, blisters, spots, abdominal pain, boils) |
| **Halli et al. 2006** [37] | *Setting*: Karnataka state, south India<br>*Population*: 1,512 FSWs aged 15–49 | **Avahan**<br>1. Membership of a sex worker collective, including self-help groups which provide information on safer sex and legal and financial support.<br>2. Contact with a peer educator<br>Extent of exposure measured by "collectivisation index" depending on whether they had been in contact with a peer educator as well as a collective<br>*Comparator*: not member of collective/lower collectivisation score (self-reported)<br>Referred to rights | 1. STI/HIV/AIDS-related knowledge<br>2. Medical attendance if symptomatic for STI<br>3. Self-reported condom-use with clients and regular partners |

*(Continued)*

**Table 1.** (Continued)

| Author and year | Population/setting (inclusion/exclusion, age, number) | Intervention (including comparator) | Outcomes |
|---|---|---|---|
| **Gangopadhyay et al. 2005** [39] | *Setting*: Kolkata, India<br>*Population*: 173 brothel-based sex workers took part in survey in Sonagachi area (central Kolkata, mean age 27) and 169 in control area (outskirts of Kolkata, mean age 32) | **Sonagachi**<br>1. Health clinic, STI treatment, information regarding STIs, condom promotion<br>2. Sex workers employed as peer educators<br>3. Empowerment of sex workers<br>*Comparator*: intervention without empowerment part<br>No explicit reference to rights but same intervention as Swendeman et al. | 1. Prevalence of STIs (clinical and lab-proven)<br>2. Use of condoms/asking clients to use condoms<br>3. Oral/anal sex<br>4. Regular STI check ups<br>5. Treatment for STI<br>6. Tested for HIV<br>7. Tried to get test result for HIV |
| **Basu et al. 2004** [40] | *Setting*: Two urban communities in the Cooch Behar district of West Bengal.<br>*Population*: 100 sex workers in each area randomly selected from the 350 sex workers in each community<br>Mean age 27 (7.04) | **Sonagachi**<br>1. Health clinic and basic STI information<br>2. Empowerment activities including raising awareness about sex worker rights<br>3. Advocacy with stakeholders<br>*Comparator*: health clinics alone (clusters randomly assigned)<br>Referred to rights | 1. Condom use (on last day worked: number of condoms used divided by number of sexual intercourse acts)<br>2. Change in condom use (between baseline and follow-up points)—Either 'adopters' (used condoms <100% of acts at baseline and 100% at follow-up or 'relapsers' (100% condom use for each act at baseline and <100% at follow up)<br>(Outcomes measured at baseline and every 5–6 months over 16 months) |
| **Jana et al. 1998** [44] **and Jana et al. 1995** [41] | *Setting*: Red light district of Sonagachi, Kolkata, West Bengal, India<br>*Population*: Baseline—random sample of 450 FSWs in the Sonagachi area (excluding "floating sex workers"—those staying in the area temporarily) stratified by professional charging rates and access to water, sanitation and other assets. 85% between 15–29 years, range 13–45 years. | **Sonagachi**<br>1. Recognition of sex workers' professional and human rights<br>2. Sexual health services including laboratory support<br>3. Education and awareness raising including of condom use<br>4. Work to improve trust with the community<br>5. Liaison with different stakeholders to produce a culture of dialogue<br>6. Health services and immunisation programmes for children of sex workers<br>7. Loans for sex workers<br>8. Counselling and social support including legal and paralegal advice<br>*Comparator*: baseline (early on in intervention)<br>Referred to rights | 1. Awareness of sexually transmitted diseases and AIDS<br>2. Condom use (% always or often)<br>3. Condom use (relative to number of sex acts performed)<br>3. VDRL seropositivity<br>4. HIV prevalence<br>(Outcomes measured in 1992, 1993, and 1995) |

FSW: Female sex worker; IQR: Interquartile range; STI: Sexually transmitted infection.

tests in the intervention group [39]. Similarly, Benzaken et al. (critical risk of bias) found increased proportions having HIV tests following intervention [34,43].

**Knowledge of STIs/STI prevention.** STI knowledge increased significantly with intervention in three studies [22,24,37], and also increased in Jana et al. who do not provide any data on whether this might have been significant [41]. Halli et al. found this association was not significant in adjusted models [37]. Two of these studies had a serious risk of bias [24,37], and two a critical risk of bias [22,41].

**Intimate partner violence.** Beattie et al. (critical risk of bias) found violence in the past year significantly reduced with the intervention [38], whilst Naved et al. (a cluster RCT at high risk of bias) found no significant association [21].

**Awareness of rights.** Swendeman et al. (serious risk of bias) reported significant increases in the proportion of FSWs feeling that sex work is valid work (and therefore they have rights related to it) [24]. Urada et al. (critical risk of bias) reported significant improvements in

Table 2. Outcomes identified in included studies.

| Author and year | Condom use | HIV | STIs other than HIV | Testing for HIV/STIs | Knowledge of STIs/prevention of STIs | Intimate partner violence (physical, sexual or emotional) | Awareness of rights | Healthcare attendance (STI clinic) | Healthcare access/interventions during pregnancy and delivery | Rights during delivery |
|---|---|---|---|---|---|---|---|---|---|---|
| Metwally et al. 2019 [27] | | | | | | | + | | | |
| Naved et al. 2018 [21] | | | | | | x | | | | |
| Ratcliffe et al. 2016 [25] | | | | | | | +*/-* | | | |
| Urada et al. 2016 [22] | +/x | | | + | + | | +/x | | | |
| Abuya et al. 2015 [42] | | | | | | | | | | +/x |
| Erausquin et al. 2012 [32] | + | | | | | | | | | |
| Guha et al. 2012 [33] | +/x | | | | | | | | | |
| Rana et al. 2012 [26] | | | | | | | | | x | |
| Deering et al. 2011 [31] | +/x | | | | | | | | | |
| Gurnani et al. 2011 [23] | | | | | | | | +* (exact number not given) | | |
| Ramesh et al. 2010 [35] | +/x | + | +/x | | | | | + | | |
| Reza-Paul et al. 2008 [36] | + | x | +/x/- | | | | | | | |
| Beattie et al. 2010 [38] | | | | | | + | | | | |
| Björkman et al. 2009 [29] | | | | | | | | x | +/x | |
| Swendeman et al. 2009 [24] | + | | | | + | | + | | | |
| Sinha 2008 [30] | | | | | | | | | +/x | |
| Benzaken et al. 2007 [34] and Benzaken et al. 2003 [43] | +/x | | | + | + | | | | | |
| Pandey et al. 2007 [28] | | | | | | | | | +/x | |
| Halli et al. 2006 [37] | +/x | | - | | +/x | | | + | | |
| Gangopadhyay et al. 2005 [39] | x | X | + | | | | | + | | |
| Basu et al. 2004 [40] | + | | | | | | | | | |
| Jana et al. 1998 [44] and Jana et al. 1995 [41] | +* | - | +* | | +* | | | | | |

No effect donated by 'x', significant (p<0.05) positive effect by '+' and negative effect by '-' (if mixed evidence found for different subgroups a mix of two or three are given).

*No significance test undertaken. When adjusted and unadjusted results are given, adjusted results are reported. STI: Sexually transmitted infection.

awareness of human rights at follow-up compared to baseline among street-based FSWs but no significant improvement for venue-based FSWs post-intervention, or for awareness of sexual and reproductive health rights (as opposed to human rights) [22]. Ratcliffe et al. (serious risk of bias) reported mixed results in terms of awareness of rights, but no significance tests were undertaken [25]. Metwally et al. (critical risk of bias) reported significant improvements in awareness of rights within a healthcare setting following intervention [27].

**Healthcare attendance (STI clinic).**   Ramesh et al. (critical risk of bias) found higher attendance rates at the project STI clinic post-intervention [35]. Gurnani et al. reported that visits to the project STI clinic increased following intervention but gave no exact figure or statistical test and was at critical risk of bias [23]. Björkman et al. (a cluster RCT with high risk of bias) found no change in numbers of family planning visits per month [29], whilst Halli et al. and Gangopadhyay et al. (both at serious risk of bias) found significantly increased attendance with intervention [37,39].

**Healthcare access/interventions during pregnancy and delivery.**   Rana et al (serious risk of bias) found no association between intervention and healthcare access during delivery, or having a skilled birth attendant present [26]. Pandey et al. (a cluster RCT with high risk of bias) found significant increases in proportions receiving prenatal examinations, supplements and tetanus vaccine but no increases in visits by the nurse-midwife [28]. Björkman et al. found no change in numbers of antenatal visits per month [29]. Sinha et al. (critical risk of bias) found increases in antenatal check-ups, giving birth in a healthcare institution and tetanus injections [30].

**Rights during delivery.**   Abuya et al. (critical risk of bias) reported that accessing rights in healthcare (not being humiliated/physically abused, having confidentiality violated, being detained, lack of consent or privacy) significantly increased straight after an educational intervention but those related to verbal aggression or abuse during delivery, examination and abandon did not significantly change [42].

**Secondary outcome: Healthcare professionals' awareness of rights.**   Ratcliffe et al. (serious risk of bias) found non-significant improved awareness among healthcare workers of the importance of confidentiality and awareness that abuse and disrespect are against human rights in maternity care but only percentages were reported (no statistical tests) [25].

## Discussion

### Findings

We found that rights-based interventions appear to be effective for certain outcomes within sexual and reproductive health, but evidence is of poor quality. Given the considerable risk of bias across all studies, the results must be interpreted with caution.

Two studies identified increased STI symptoms/diagnosis following intervention, but it is unclear whether this reflects a true increase. For instance, there was an increase in Herpes Simplex Virus-2 diagnosis in Reza-Paul et al.'s study post-intervention but the prevalence of other STIs decreased [36]. Halli et al. found increased STI symptoms among those with higher self-reported intervention exposure, although this could relate to a greater awareness of symptoms among those exposed to the intervention [37].

In keeping with a previous systematic review (mentioned above), which looked at interventions aiming to promote awareness of rights in order to increase use of maternity services [12], our study found mixed results in terms of the effect of interventions on healthcare access during pregnancy and delivery; of the four studies in our systematic review reporting this outcome, three were included in both reviews. The additional study reported no significant effect of intervention on whether women delivered in a healthcare setting (versus at home), whether

a skilled birth attendant was present or whether a clean home delivery kit was used [26]. This relative lack of evidence for rights-based approaches in maternity settings is disappointing, and further studies are urgently needed, particularly given the recommendation for rights-based approaches in the OHCHR Technical Guidance [11].

In contrast to Stangl et al.'s systematic review of the impact of human rights programmes on HIV-related outcomes which found them to be largely effective [13], our systematic review showed mixed results: one study found a reduction in HIV prevalence [35], one reported no change [36] and one reported a significant increase [44] (although all were at critical risk of bias). Interestingly, only one study overlapped between the two systematic reviews, largely due to a different interpretation of a rights-based approach [23]. The overlapping study was limited to outcomes related to news reporting and intervention exposure/service use (not HIV-specific).

## Population

Over half of the twenty-four studies included were undertaken among FSWs in India [23,24,31–33,35–41,44]. These studies were evaluating aspects of either the India AIDS initiative [23,31–33,35–38] or the Sonagachi Project [24,39–41,44]. The remaining eleven studies were undertaken across a fairly wide geographical footprint (India, Bangladesh, Philippines, Tanzania, India, Nepal, Kenya, Brazil and Egypt). Only seven studies were undertaken among a population that was not made up of FSWs [21,25,27–30,42], most of which were among women who were pregnant or had recently given birth. There remains a paucity of evidence on rights-based approaches among those who do not engage in sex work, and in non-maternity settings.

## Study identification

Many studies identified in this systematic review were not identified from database searching. This is in keeping with a systematic review mentioned above where one of the four studies included was identified from searches and the other three from discussions with experts [12]. This may be because the word 'rights' is not often mentioned in key words, abstracts or titles [12]. We found that even if an intervention was described as related to 'rights' in one paper, another paper evaluating the same intervention (but evaluating different outcomes) did not necessarily refer to the same intervention as related to 'rights'. For example, Reza-Paul et al. described a community mobilisation and outreach intervention as using a "rights-based approach" [36], whilst Ramesh et al.'s study did not mention 'rights' despite evaluating the same intervention [35].

Similarly, another systematic review found that despite the studies included using what authors defined as human rights-based approaches, many studies did not explicitly refer to the protection and promotion of human rights in reference to the intervention being evaluated [13]. Thus, intervention descriptions and whether interventions are 'rights-based' is subjective, and some relevant studies may have been missed in this systematic review. Given the importance of women achieving their sexual and reproductive health rights [6,11], future studies should ensure they are explicit in their use of a rights-based approach or at least refer to 'rights' in relation to the intervention if they are promoted so that the available evidence can be appropriately evaluated.

## Outcomes not identified from included studies

Not all important sexual and reproductive health outcomes were identified in the papers included. No results were identified regarding legal action or convictions related to gender-

based violence; this is consistent with a previous systematic review of systematic reviews of gender-based violence prevention among adolescent girls in low-income countries which found a paucity of evidence [45]. Gurnani et al. described 4,600 rights violations towards FSWs reported to the police during the study period, and crisis management teams supported 92% of these, but it was unclear whether this was a change from previously [23]. Ratcliffe et al. found that 10% of women who attended their Open Birth Days intervention filed a complaint to the hospital regarding their treatment following the intervention, compared to no women before the intervention, but they did not refer to legal actions or convictions [25]. Additionally, outcomes related to perinatal mental health, unintended pregnancy or abortion, delivery outcomes, female genital mutilation or mental wellbeing were not identified in any included papers.

## Limitations

All included studies used complex interventions, usually involving multiple components with rights being some part of it. There was no consistent approach to incorporating rights into programmes. Even in situations where studies found the interventions effective, it was unclear whether or not this was due to the rights-based aspect. Detail regarding interventions in many papers was limited and it was often unclear how the rights-based element was incorporated, exacerbating the challenge in identifying whether a rights-based approach is effective.

Our inclusion criteria were based on the author recognising that the approach was rights-based and making that explicit. Not having our own definition of what we would consider a rights-based approach means that there may have been studies that were not included, but which examined interventions that did have a rights-based approach. Note that several studies we included did not mention that the interventions studied were rights-based, but because they examined the same intervention that had been described that way in other publications, we included them. This suggests there may be studies our systematic review has missed. Similarly, it is possible that some of the studies included would not have met a strict definition of what a rights-based approach should entail. Reassuringly, none of the included studies were obviously in this category.

Since this systematic review was limited to females, some studies were excluded due to outcomes not being separated by gender, and some outcomes within included studies were also excluded for this reason. For example, Rana et al. included relevant outcomes that were not separated by gender (even if it may be assumed that most people answering the question were women): data were available on knowledge of circumstances when abortion is legal or illegal, HIV and STIs, and contraceptive methods but as these data were not available for women only [26], they were not included in this systematic review.

As discussed, the quality of studies overall was poor. Most studies used an uncontrolled before-and-after study design. These are by nature prone to bias as there is no way of knowing what might have happened without the intervention and what else occurred in the time period which might actually have been responsible for any findings: they are therefore at high risk of confounding [46]. Moreover, some studies did not undertake any statistical analysis of results nor explain why analysis was not undertaken [23,25,41,44]. We did not undertake a meta-analysis due to the heterogeneity of populations, interventions and outcomes. This limited us in undertaking a rigorous estimate of effectiveness.

## Conclusions

This is the first systematic review to evaluate the evidence on rights-based approaches to sexual and reproductive health including maternity, gender-based violence, sexual health and HIV.

We undertook a comprehensive search of the literature from three relevant leading electronic databases, as well as reference searching of included studies and relevant systematic reviews. By synthesising the evidence base, we have illuminated the evidence gaps and what should be done in future research to improve the quality of the evidence base. Future research should be more explicit about the use of rights-based approaches and specify more precisely what distinguishes a rights-based approach from others, in a way that would allow evidence to be gathered. Moreover, evidence is needed across a diverse range of populations (not limited to FSWs and maternity settings). Given that rights-based approaches are recommended widely [6,47], high quality (ideally cluster randomised) controlled studies need to be undertaken urgently to determine whether rights-based approaches to sexual and reproductive health are effective in LMICs.

## Supporting information

**S1 Table. Risk of bias in cluster randomised controlled trials.**
(DOCX)

**S2 Table. Risk of bias in non-randomised interventional studies.**
(DOCX)

**S3 Table. Risk of bias in uncontrolled before-and-after studies.**
(DOCX)

**S1 File. Search strategy used for MEDLINE.**
(DOC)

**S2 File. PRISMA checklist.**
(DOCX)

## Acknowledgments

The authors wish to thank Samantha Johnson, Academic Librarian at the University of Warwick, for providing feedback on the search strategy, and Dr Yen-Fu Chen, Associate Professor at the University of Warwick, for support with the translation of a full text study in Mandarin.

## Author Contributions

**Conceptualization:** Majel McGranahan, Sharifah Sekalala, Oyinlola Oyebode.

**Data curation:** Majel McGranahan, Joselyn Nakyeyune, Christopher Baguma, Nakibuuka Noor Musisi, Derrick Nsibirwa, Oyinlola Oyebode.

**Funding acquisition:** Majel McGranahan, Sharifah Sekalala, Oyinlola Oyebode.

**Investigation:** Majel McGranahan, Joselyn Nakyeyune, Christopher Baguma, Nakibuuka Noor Musisi, Derrick Nsibirwa, Oyinlola Oyebode.

**Methodology:** Majel McGranahan, Oyinlola Oyebode.

**Supervision:** Oyinlola Oyebode.

**Writing – original draft:** Majel McGranahan.

**Writing – review & editing:** Majel McGranahan, Joselyn Nakyeyune, Christopher Baguma, Nakibuuka Noor Musisi, Derrick Nsibirwa, Sharifah Sekalala, Oyinlola Oyebode.

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
