## [Decision Letter · Decision Letter 0]

13 Jan 2021

PONE-D-20-29838

Rights based approaches to sexual and reproductive health in low and middle-income countries: a systematic review

PLOS ONE

Dear Dr. McGranahan,

Thank you for submitting your manuscript to PLOS ONE. After careful consideration, we feel that it has merit but does not fully meet PLOS ONE’s publication criteria as it currently stands. Therefore, we invite you to submit a revised version of the manuscript that addresses the points raised during the review process.

We look forward to receiving your revised manuscript.

Kind regards,

Florian Fischer

Academic Editor

PLOS ONE

Journal Requirements:

2. Please attach a Supplemental file of the results of the individual components of the quality assessment, not just the overall score, for each study included. Thank you.

Reviewers' comments:

Reviewer's Responses to Questions

**Comments to the Author**

1. Is the manuscript technically sound, and do the data support the conclusions?

Reviewer #1: Yes

Reviewer #2: Partly

2. Has the statistical analysis been performed appropriately and rigorously? 

Reviewer #1: Yes

Reviewer #2: No

3. Have the authors made all data underlying the findings in their manuscript fully available?

Reviewer #1: Yes

Reviewer #2: Yes

4. Is the manuscript presented in an intelligible fashion and written in standard English?

Reviewer #1: Yes

Reviewer #2: Yes

5. Review Comments to the Author

Reviewer #1: The reviewed paper focused on rights-based approaches (from a rights holder perspective) to sexual and reproductive health (SRH) of adolescent girls and women in low and middle-income countries (LMICs). The authors found that right-based approaches seemed to be effective in the context of SHR. However, the examined approaches lacked evidence.

In total, I would suggest this paper to be published. The researched topic is of relevance to Global and Public Health. Further, the paper is well written and the different sections are coherent between each other. However, I would suggest minor changes, which will be stated in the following paragraphs, stating the chapters of the paper.

The abstract was written in a logical order and reflects the relevant aspects of the article. Adjustments: 1.) I would suggest stating the concrete age for the population, rather than refereeing to just words. That would be a more precise way to reflect the study population included in this study.

The introduction states the relevance of the topic rather well and is coherent in that the authors describe all relevant aspects regarding the current situation of sexual and reproductive health in LMICs and right-based approaches. Further, the authors create a connection between the two topics by describing the current state of the art. Hence, they validate why this review is of importance. Adjustments: none

The methods are written with clear language and in a logical order. Adjustments: 1.) I would recommend that the objectives be made more precise. The added value of the review should be made clear. 2.) While the appendix contains the PRISMA flow diagram and checklist, it was not indicated in the Methods section that the authors followed the PRISMA statement. I would suggest stating this at the beginning of the chapter. 3.) By naming the search terms in the appendix, it is transparent how the authors obtained their results. Nevertheless, I would recommend, to briefly mention which subject areas are included in the search algorithm in the methods. 4.) Further, the authors describe the inclusion criteria in detail; however, do not this in a similar way for the exclusion criteria. To be more precise, I would empathize doing this. 5.) The inclusion criteria, also, include a bullet point as exclusion criteria, which is a little bit confusing to me. 6.) As already mentioned in the abstract section, I would suggest stating the age of the study population specifically.

The results are clearly arranged and correspond to the tables presented. The tables support the text and are comprehensible. Overall, the results support the conclusion of the paper. Adjustments: 1.) Table 1. should be reviewed to coherent writing between the columns. For instance, in the column ‘population/setting (…)’ Metwalley et al. and Benzaken et al., the setting is stated directly (e.g. Egypt) and Naved et al. start with the heading (e.g. the setting). 2.) For table 3. and 4., it would be helpful to have the tool used for categorizing the risk of bias, as additional information underneath. That would help the reader to remember the underlying basis on which the categorization was made.

The discussion summarizes the most relevant results once again and puts them into a different context. Further, the authors describe the limits of their research. Adjustments: I have one question to ask. Why did the authors not use their definition of right-based approaches, to include a broader range of studies? I think that this question should be answered as well.

The conclusion of this paper is, as already said, justified by the results. Further, it is short and precise. Hence, no adjustments are necessary.

Reviewer #2: Reviewer comments for “Rights based approaches to sexual and reproductive health in low and middle-income countries: a systematic review”

Introduction

More explanation of the decision to exclude duty-bearers type approaches in this review is warranted.

The authors should present a definition for sexual and reproductive health. This will better highlight the gap in the literature that they are claiming the review fills. For example, does the authors’ definition of sexual and reproductive health include safe access to abortion?

Methods

The methods states that there were no language restrictions. If that was the case, how many languages were include and who conducted the review of published studies in languages other than English?

The authors should describe their criteria for “rights-based approaches” further. Some of the included studies seem to be educational interventions so some distinction should be provided for when educational interventions are included vs. excluded.

It seems abortion was left off the acceptable outcomes list. One of the central arguments at the intersection of rights and sexual and reproductive health is women’s right to safe and equitable abortion care. Thus, this review is incomplete with regards to sexual and reproductive health and it is an oversight to leave out the abortion literature in its entirety.

Other outcomes to consider are impacts on OBGYNs and advanced practice providers who provide SRH care specifically.

Results

Table 2 is not an effective way to present this information. A table that is similar to a “sample description” of all the included papers would be more effective and could include the distribution of studies across each of the outcomes, the study design, etc.

There should be an additional results table that aggregates information across the studies in a manner that is easy for readers to understand. If one is to quickly glance at the tables, they should be able to garner most of the information about this set of studies. Perhaps a table that further aggregates the information on types of “rights-based approaches” would be suitable.

Tables 3 – 5 should be moved to the Appendices. The paragraph of results about the bias assessment can remain in the main text of the paper. Otherwise, the bulk of the tables are related to bias.

Discussion

There are several grammatical errors in the discussion section and it should be revised with attention to detail.

The fact that there is a similar review that focuses on maternity care is concerning, particularly, since 3 of the same studies are found in both reviews and there are sexual and reproductive health outcomes that are left out of this review.

Another point of concern is that only 7 studies focus on non-FSW populations. Should this review be limited then to actor-based rights-based approaches to FSW SRH? It is not convincing to make any conclusions about the health of other groups based on seven studies, particularly, given the amount of bias reported in the studies as a whole.

The approach undertaken in this review cannot be easily replicable. This is because most of the studies came from non-systematic approaches. But a large part of systematic reviews is the ability to reproduce the findings.

Most of the discussion focuses on the limitations of this study and approach. Thus, it remains unclear what this review contributes to the existing literature. The limitations seem to overwhelm any merit that this work could have. The evidence to support the existence of such a systematic review is simply not there.

6. PLOS authors have the option to publish the peer review history of their article (what does this mean?). If published, this will include your full peer review and any attached files.

Reviewer #1: No

Reviewer #2: No

---

## [Author Response · Author response to Decision Letter 0]

3 Feb 2021

Dear Editorial Team,

Re: Manuscript ID PONE-D-20-29838 – “Rights based approaches to sexual and reproductive health in low and middle-income countries: a systematic review”

We thank the editorial team and reviewers for taking the time to read our manuscript and for your helpful comments. We feel that by addressing your suggested changes, the paper is much improved, and we hope that you will agree.

Please find attached (in the document entitled 'Response to Reviewers') a table outlining our response to each issue raised. 

Best wishes,

Dr Majel McGranahan, on behalf of the review authors

---

## [Decision Letter · Decision Letter 1]

10 Mar 2021

PONE-D-20-29838R1

Rights based approaches to sexual and reproductive health in low and middle-income countries: a systematic review

PLOS ONE

Dear Dr. McGranahan,

Thank you for submitting your manuscript to PLOS ONE. After careful consideration, we feel that it has merit but does not fully meet PLOS ONE’s publication criteria as it currently stands. Therefore, we invite you to submit a revised version of the manuscript that addresses the points raised during the review process.

Your manuscript has now been assessed by a third reviewer, who has not been involved in the first found of review. Please adjust your manuscript according to the recommendations raised by Reviewer 3 (see below).

We look forward to receiving your revised manuscript.

Kind regards,

Florian Fischer

Academic Editor

PLOS ONE

Journal Requirements:

Reviewers' comments:

Reviewer's Responses to Questions

**Comments to the Author**

1. If the authors have adequately addressed your comments raised in a previous round of review and you feel that this manuscript is now acceptable for publication, you may indicate that here to bypass the “Comments to the Author” section, enter your conflict of interest statement in the “Confidential to Editor” section, and submit your "Accept" recommendation.

Reviewer #3: (No Response)

2. Is the manuscript technically sound, and do the data support the conclusions?

Reviewer #3: Partly

3. Has the statistical analysis been performed appropriately and rigorously? 

Reviewer #3: N/A

4. Have the authors made all data underlying the findings in their manuscript fully available?

Reviewer #3: Yes

5. Is the manuscript presented in an intelligible fashion and written in standard English?

Reviewer #3: Yes

6. Review Comments to the Author

Reviewer #3: The premise of this article is good – what is the evidence related to taking a human rights-based approach to SRHR. The article does not actually do this though, since a “human rights-based approach” seems to anything the authors of the studies included in the review said it was, including if the author said the intervention related to rights. I appreciate that the authors of the review said they were concerned that having their own definition might exclude some studies, but at the same time, the absence of a common definition of a rights-based approach (or what it means to be “related to rights), means that the reader doesn’t learn much about taking a rights-based approach to SRH – and how that differs from just labeling an intervention part of SRHR. If they simply said their intervention addressed rights, how can the authors of this review say that was that sufficient to include as a rights-based approach?

People have rights that they can claim, and rights standards and principles can be incorporated into programming. For example, individuals and couples have the right to decide freely and responsibly the number and spacing of their children, and they have the right to information and services to act on that right, with equity and free of discrimination and coercion. For contraceptive information and services, the rights principles and standards that need to be incorporated into programs are: availability, accessibility, acceptability and quality (AAAQ of the right to the highest attainable standard of health), informed decision-making, privacy and confidentiality, non-discrimination, participation and accountability (WHO, 2014, Ensuring Human Rights in Contraceptive Information and Services). The authors should describe what about “rights” and “rights-based approach” is noted in the studies included in the review. This would be an important contribution to the literature – what does it mean to take a rights-based approach to programming – in various areas of health.

The authors note the SDG – Goal 3, but they equate it with SRHR, whereas Goal 3 is not ensuring that everyone can access SRHR – it is “Ensure healthy lives and promote well–being for all at all ages.” Targets under Goal 3 include reducing maternal mortality; ending preventable child deaths; ending or reducing AIDS other diseases; universal health coverage, affordable essential medicines, sexual and reproductive health care; vaccine research, and access to medicines. SRHR is measured in the SDG through two indicators – “Proportion of women of reproductive age (aged 15–49 years) who have their need for family planning satisfied with modern methods,” and the adolescent birth rate. https://sdg-tracker.org/good-health. So, according to the SDG, maternal health, HIV and GBV, which are the services covered in the review by the authors, are not technically part of SRH. That doesn’t negate the review – but the authors should not use the shorthand of “SRHR” to cover the topics they have included in the review – or they should say that they started with a broad list of SRH outcomes but only found relevant studies associated with a few of the SRH outcomes.

The first sentence of the abstract: “By 2030, the Sustainable Development Goals aim to achieve human rights for all” is not accurate statement – the SDG are grounded in human rights, but the goals are not stated as “achieving human rights for all.” Suggest using the term “grounded in” – instead. https://www.ohchr.org/Documents/Issues/MDGs/Post2015/SDG_HR_Table.pdf

The results part of the abstract notes that: rights-based interventions were effective for most included outcomes, but evidence was of poor quality. My reading of the table with results is that the findings were mixed. Also, the limitations section of the paper indicates that it was often not possible to ascertain if the rights part of the intervention made the difference to outcomes – to me the bigger message from this review is that we don’t have common/consistent ways of incorporating rights into programming, and studies to assess the effect of taking a rights-based approach (or to address rights) are of low quality and more, better designed studies, with clear theories of change, are urgently needed. The authors make this point in the conclusion.

The final sentence of the abstract: “Considerable risk of bias in all studies means results must be interpreted with caution. High-quality controlled studies are needed urgently in this area.” What is meant by “bias” in this sentence?

Introduction, lines 68-72. The distinction between rights vs. needs is likely to go over most readers heads – is it necessary? If so, it should be explained.

Page 4, lines 76-79. The authors imply that UNFPA’s take on a human rights-based approach only has two components – those related to duty bearers and rights-holders. The reference (https://www.unfpa.org/human-rights-based-approach) includes a lot more than those 2 components. I suggest a more robust discussion of what a rights-based approach is, including reference to the UN Common Understanding of HRBA.

Page 5, lines 83-88 focus on empowering women to claim their rights. That is important, but again, is not all there is to a human rights-based approach. What needs to be in place for women to claim their rights?

Page 11, Lines 203-204. Given the list of outcomes searched (shown on lines 122-123 - contraception, pregnancy, STIs, awareness of rights, violence and mental health), it would be good here to list the outcomes included in the 24 studies – this will help the reader understand why only some outcomes are included in the results section of the paper.

It is not surprising that half of the studies are related to HIV – human rights were more actively and explicitly promoted as part of HIV programming than in other areas covered in the review.

Table 1 – I urge the authors to group studies rather than just listing them in chronological order. What about all of the HIV studies together, all of the maternal health studies together, etc. That will help readers look for patterns among the studies addressing each outcome.

The title of Table 2 is misleading – the review did not assess the “effectiveness of a rights-based approach” – it included studies in which rights or rights-based were mentioned. As the authors say in the paper, there is no way of knowing what effect the “rights” or “rights-based” part had in the outcome.

Page 23 – discussion of the outcomes. The authors note that of the studies measuring condom use as an outcome, “all of these studies were at serious, high or critical risk of bias.” Please explain in the methods section what is meant by “bias.” What is on pages 25-26, lines 322-327, should be moved into the methodology – and bias better explained that simply referring to supplemental tables (it is good to keep the supplemental tables, but readers need to better understand what bias is and how it was assessed in this review.

7. PLOS authors have the option to publish the peer review history of their article (what does this mean?). If published, this will include your full peer review and any attached files.

Reviewer #3: **Yes: **Karen Hardee

---

## [Author Response · Author response to Decision Letter 1]

15 Apr 2021

Dear Editorial Team,

Re: Manuscript ID PONE-D-20-29838 – “Rights based approaches to sexual and reproductive health in low and middle-income countries: a systematic review”

We thank the third reviewer, Dr Karen Hardee, for taking the time to read our manuscript and for your helpful comments. We feel that by addressing your suggested changes, the paper is much improved, and we hope that you will agree.

Please find attached (in the document entitled 'Response to Reviewers') a table outlining our response to each comment.

With best wishes,

Dr Majel McGranahan, on behalf of the review authors

---

## [Decision Letter · Decision Letter 2]

19 Apr 2021

Rights based approaches to sexual and reproductive health in low and middle-income countries: a systematic review

PONE-D-20-29838R2

Dear Dr. McGranahan,

We’re pleased to inform you that your manuscript has been judged scientifically suitable for publication and will be formally accepted for publication once it meets all outstanding technical requirements.

Kind regards,

Florian Fischer

Academic Editor

PLOS ONE

Additional Editor Comments (optional):

Reviewers' comments:

Reviewer's Responses to Questions

**Comments to the Author**

1. If the authors have adequately addressed your comments raised in a previous round of review and you feel that this manuscript is now acceptable for publication, you may indicate that here to bypass the “Comments to the Author” section, enter your conflict of interest statement in the “Confidential to Editor” section, and submit your "Accept" recommendation.

Reviewer #3: All comments have been addressed

2. Is the manuscript technically sound, and do the data support the conclusions?

Reviewer #3: Yes

3. Has the statistical analysis been performed appropriately and rigorously? 

Reviewer #3: N/A

4. Have the authors made all data underlying the findings in their manuscript fully available?

Reviewer #3: Yes

5. Is the manuscript presented in an intelligible fashion and written in standard English?

Reviewer #3: Yes

6. Review Comments to the Author

Reviewer #3: (No Response)

7. PLOS authors have the option to publish the peer review history of their article (what does this mean?). If published, this will include your full peer review and any attached files.

Reviewer #3: **Yes: **Karen Hardee

---

## [Editor Report · Acceptance letter]

21 Apr 2021

PONE-D-20-29838R2 

Rights based approaches to sexual and reproductive health in low and middle-income countries: a systematic review 

Dear Dr. McGranahan:

I'm pleased to inform you that your manuscript has been deemed suitable for publication in PLOS ONE. Congratulations! Your manuscript is now with our production department. 

Kind regards, 

on behalf of

Dr. Florian Fischer 

Academic Editor

PLOS ONE